# NOT ALL ROLLOUTS ARE USEFUL: DOWN-SAMPLING ROLLOUTS IN LLM REINFORCEMENT LEARNING

## ABSTRACT

Reinforcement learning with verifiable rewards (RLVR) has emerged as the leading approach for enhancing reasoning capabilities in large language models. However, it faces a fundamental compute and memory asymmetry: rollout generation is embarrassingly parallel and memory-light, whereas policy updates are communication-heavy and memory-intensive. To address this, we introduce **PODS** (**P**olicy **O**ptimization with **D**own-**S**ampling), which decouples rollout generation from policy updates by training only on a strategically selected subset of rollouts, maintaining learning quality while dramatically reducing update costs. We propose a principled subset selection criterion—*max-variance down-sampling*—that maximizes reward diversity, and provide an efficient $O(n \log n)$ implementation. Empirically, Group Relative Policy Optimization (GRPO) with PODS achieves the peak test accuracy of vanilla GRPO at least $\mathbf{1.7\times}$ **faster** across the different reasoning benchmarks and hardware configurations we tested.

## 1 INTRODUCTION

Reinforcement learning with verifiable rewards (RLVR) has driven recent breakthroughs in solving math problems, code generation, and general reasoning for large language models (LLMs) (Jaech et al., 2024; Ziegler et al., 2019; Ouyang et al., 2022; Stiennon et al., 2020). RLVR algorithms such as Proximal Policy Optimization (PPO) (Schulman et al., 2017) and Group Relative Policy Optimization (GRPO) (Shao et al., 2024) share a two-phase structure: an *inference phase*, which generates rollouts given a prompt, and a *policy-update phase*, which updates the model parameters using the rewards calculated on those rollouts.

These two phases place different computational demands on the hardware. Inference is embarrassingly parallel and relatively memory-light, enabling modern accelerators to produce thousands of rollouts concurrently. Although generating a single rollout may have high latency due to autoregressive decoding, batching rollouts amortizes the per-token latency and yields higher throughput. Policy updates, on the other hand, scale poorly with batch size: they are memory- and communication-intensive, requiring full-precision optimizer states and cross-device synchronization of gradients and parameters. This asymmetry creates a fundamental bottleneck: systems must either throttle inference (underutilizing compute) or resort to memory-saving techniques like gradient accumulation (increasing communication overhead and policy update latency), both of which hurt training efficiency. Fig. 1 provides empirical evidence for this computational asymmetry.

We address this bottleneck through a key observation: *not all rollouts contribute equally to model improvement*. Beyond a certain scale, additional rollouts provide diminishing returns and can even degrade learning signals through redundant information. This suggests a natural solution: generate large batches of rollouts during the scalable inference phase, but train selectively on only the most informative subset during the policy update phase, avoiding the latency overhead of memory-saving techniques. We formalize this idea in **PODS** (**P**olicy **O**ptimization with **D**own-**S**ampling). As illustrated in Fig. 2, PODS maximizes hardware utilization by generating $n$ rollouts per prompt but updating on only $m < n$ informative samples selected by a principled down-sampling rule.

Within the PODS framework, we introduce *max-variance down-sampling*, a principled criterion that selects the subset of rollouts with the greatest reward variance of the selected subset, thereby preserving strong contrastive signals. We show that the resulting combinatorial problem can be solved in $O(n \log n)$ time and, in the common binary-reward setting, reduces to picking the $m/2$

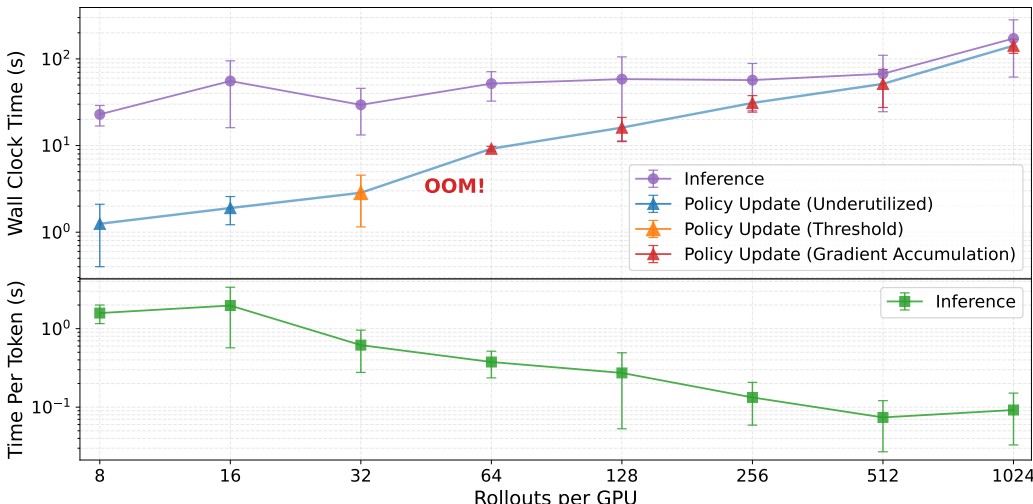

Figure 1: **Inference scales efficiently while policy updates become memory-bound in RLVR.** Empirical timing breakdown when fine-tuning Qwen2.5-3B-Instruct on GSM8K using 8 A100-80GB GPUs with varying rollouts per GPU. **Top:** Total wall-clock time per iteration. Policy updates hit memory limits after 32 rollouts per GPU (OOM beyond this point), requiring gradient accumulation that dramatically slows training. **Bottom:** Per-token inference time decreases $21\times$ through batching (from 8 to 512 rollouts), saturating beyond 512. This demonstrates the core asymmetry that PODS exploits: inference parallelizes efficiently while policy updates become memory-bound.

highest-reward and $m/2$ lowest-reward rollouts. We evaluate PODS with GRPO on GSM8K (Cobbe et al., 2021) and MATH (Hendrycks et al., 2021) across multiple model and hardware configurations, demonstrating that it achieves the peak test accuracy of baseline GRPO at least $1.7\times$ **faster**.

## 2 RELATED WORK

**Reinforcement learning for LLM reasoning.** Reinforcement learning has emerged as a powerful paradigm for enhancing the reasoning capabilities of LLMs across math, coding, and problem-solving domains (Jaech et al., 2024; Shao et al., 2024; Kazemnejad et al., 2024). Although classical algorithms such as Proximal Policy Optimization (PPO) (Schulman et al., 2017) laid the foundation, recent work has tailored them specifically to language models. Group Relative Policy Optimization (GRPO) (Shao et al., 2024) has gained prominence for reasoning tasks because of its implementation simplicity, competitive performance relative to PPO, and lack of a separate critic network. OpenAI o1 (OpenAI, 2024) and DeepSeek R1 (Guo et al., 2025), which used large-scale RL, have sparked interest in reasoning-focused RL methods (Chen et al., 2025; Hu et al., 2025; Hu, 2025; Cui et al., 2025). Meanwhile, value-based approaches like PPO remain central (Yuan et al., 2025a;b), alongside complementary techniques such as Monte Carlo Tree Search (Gao et al., 2024; Xie et al., 2024) and multi-agent methods (FAIR et al., 2022). A recent line of work has also explored data selection for improving RL methods for LLM training. Specifically, prompt selection and filtering has gained significant attention from works such as DAPO (Yu et al., 2025), SRPO (Zhang et al., 2025), Reinforce-Rej (Xiong et al., 2025), Polaris An et al. (2025) and VAPO (Yuan et al., 2025a). Our method advances this line of work by focusing on down-sampling rollouts within each prompt, instead of selecting or filtering prompts themselves. By tackling this computational-efficiency bottleneck, our approach complements existing methods and can be combined with them to further improve reasoning performance.

**Down-sampling and data selection.** The scale of modern machine learning necessitates effective data management strategies, particularly as datasets grow larger, noisier, and more imbalanced. Training on the full dataset can be prohibitively expensive, motivating sophisticated data-selection and down-sampling methods. Such techniques succeed across diverse settings—from theoretical results

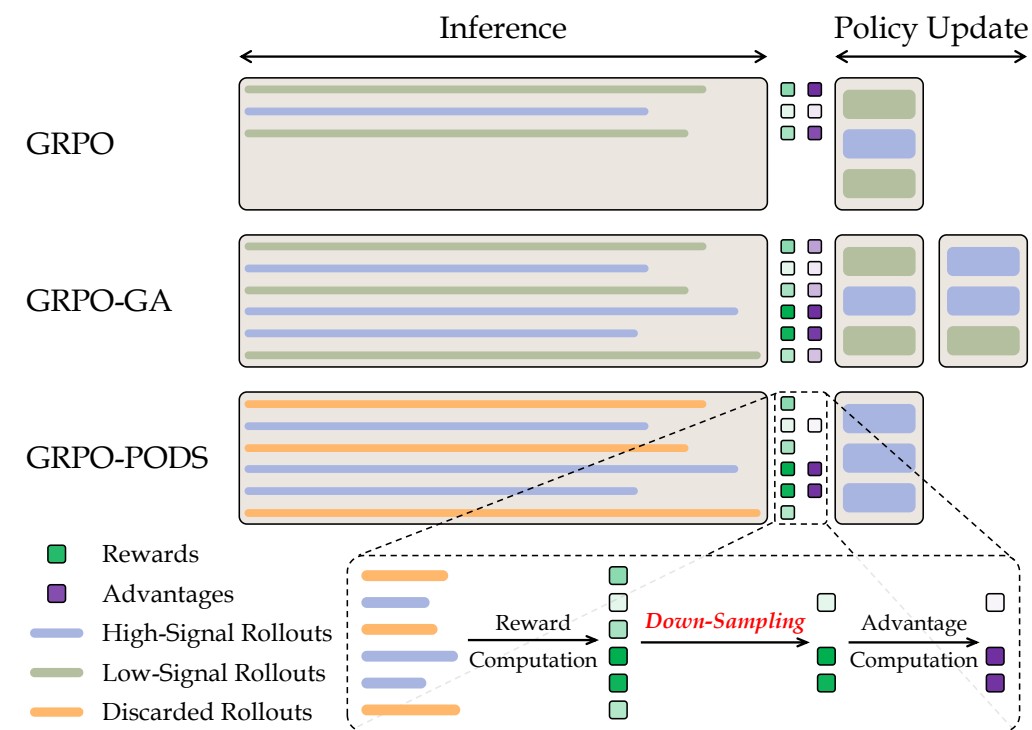

Figure 2: Visualization of three training strategies: vanilla GRPO, GRPO with gradient accumulation (GRPO-GA), and GRPO with PODS (GRPO-PODS). Vanilla GRPO generates $n$ rollouts and trains on all of them, leaving inference hardware underutilized. GRPO-GA alleviates this issue with memory-saving techniques such as gradient accumulation, but at the cost of more sequential steps in the policy-update phase. In contrast, GRPO-PODS also generates $n$ rollouts but trains on only $m$ carefully selected examples, maximizing inference utilization, avoiding gradient-accumulation overhead, and providing a cleaner learning signal that yields better final performance.

in clustering (Har-Peled & Mazumdar, 2004), regression (Li et al., 2013; Rudelson & Vershynin, 2007; Clarkson, 2010) to practical systems in speech recognition (Liu et al., 2015; Wei et al., 2014) and computer vision (Kaushal et al., 2019; Bankes et al., 2024). In reinforcement learning, prioritized experience replay (Schaul et al., 2015) and related methods (Hou et al., 2017; Saglam et al., 2023; Cusumano-Towner et al., 2025) highlight the value of selective sampling from experience buffers. More recently, careful data selection has become central to foundation-model training (Goyal et al., 2024; Schuhmann et al., 2021; Gadre et al., 2023) and emerging applications such as computational advertising (Bei et al., 2023; Gravin et al., 2024). Yet, to our knowledge, we are the first to apply principled down-sampling to the rollout-generation stage of LLM reinforcement learning, mitigating a key computational bottleneck while strengthening the learning signal.

## 3 DOWN-SAMPLING ROLLOUTS IN GRPO

In this section, we present our approach to resolving the computational asymmetry between inference and policy updates in LLM reinforcement learning. We first review the original GRPO algorithm in Section 3.1, highlighting its *structural* components and computational demands. Next, in Section 3.2, we introduce the **PODS** (**P**olicy **O**ptimization with **D**own-**S**ampling) framework, which strategically selects informative rollouts to maximize hardware utilization during both inference and policy-update phases. In Section 3.3, we develop a principled *max-variance down-sampling* method that preserves strong contrastive signals, justified by Razin et al. (2025), by retaining only rollouts from the extremes of the reward spectrum. We show that this method admits an elegant, $O(n \log n)$ solution, making it practical for real-world deployment. Overall, our framework retains the advantages of GRPO while boosting computational and memory efficiency across diverse hardware setups.

### 3.1 PRELIMINARIES

Group Relative Policy Optimization (GRPO) (Shao et al., 2024) is a reinforcement-learning algorithm intended to enhance the reasoning capabilities of large language models (LLMs), particularly within the RLVR setting. Each GRPO training step follows a structured, two-phase process, described below.

**Inference phase.** Let $\pi_\theta$ denote the policy parameterized by $\theta$, which defines a distribution over next-token probabilities given the previous tokens in a sequence. Given a single input prompt $p$ (e.g., a math problem), GRPO first generates a group of $n$ rollouts $\mathbf{o} = (o_1, o_2, \ldots, o_n)$ by autoregressively sampling from $\pi_\theta$. Each rollout is a complete token sequence excluding the prompt, representing a possible solution. Each rollout is then evaluated using a reward model $r_i = R(o_i)$, which scores the quality and correctness of the corresponding output $o_i$. This yields a reward vector $\mathbf{r} = (r_1, r_2, \ldots, r_n)$. From these rewards, we compute normalized advantage estimates: $a_i = (r_i - \mu)/\sigma$, where $\mu$ and $\sigma$ are the mean and standard deviation of the rewards respectively.

**Policy update phase.** After computing the advantages, the policy is updated by optimizing the GRPO objective $L_{\mathrm{GRPO}}(\theta)$. Specifically, for each rollout $o_i$ with advantage $a_i$, we compute a loss for each token position $t$, and then average over all tokens and rollouts:

$$L_{\mathrm{GRPO}}(\theta) = \frac{1}{n} \sum_{i=1}^{n} \frac{1}{|o_i|} \sum_{t=1}^{|o_i|}$$

$$\min \left[ \frac{\pi_\theta(o_{i,t} \mid p, o_{i,<t})}{\pi_{\theta_{\mathrm{fixed}}}(o_{i,t} \mid p, o_{i,<t})} \cdot a_i, \ \mathrm{clip} \left( \frac{\pi_\theta(o_{i,t} \mid p, o_{i,<t})}{\pi_{\theta_{\mathrm{fixed}}}(o_{i,t} \mid p, o_{i,<t})}, 1 - \epsilon, 1 + \epsilon \right) \cdot a_i \right].$$

where $|o_i|$ is the number of tokens in $o_i$ and $\pi_{\theta_{\mathrm{fixed}}}$ is a frozen copy of the policy used for importance weighting. This asymmetric loss embodies the *slow to adopt, quick to abandon* learning principle—limiting how aggressively the policy increases probabilities for tokens in high-reward rollouts while allowing more substantial reductions for low-reward sequences.

### 3.2 PODS FRAMEWORK

We propose to *decouple the inference and training phases* in GRPO. Rather than updating on every generated rollout, PODS first produces $n$ rollouts in parallel and then trains on only a smaller subset of size $m < n$ selected by a down-sampling rule $D$. This strategy exploits parallelism during inference while substantially reducing the communication and memory costs of the subsequent policy update.

**Definition 3.1** (Down-sampling rule). $D(\mathbf{o}, \mathbf{r}; m)$ *is a function that takes as inputs $n$ rollouts $\mathbf{o} = (o_1, o_2, \ldots, o_n)$, their corresponding rewards $\mathbf{r} = (r_1, r_2, \ldots, r_n)$, and the update size $m$. It outputs a subset of indices $S \subseteq \{1, 2, \ldots, n\}$, where $|S| = m$, indicating which rollouts to retain for the policy update phase.*

Given a selected subset of indices $S$, we compute the advantage estimates using only the selected rollouts: $a_{S,i} = (r_i - \mu_S)/\sigma_S$, where $\mu_S$ and $\sigma_S$ are the mean and standard deviation of the rewards in the selected subset. The GRPO-PODS objective then becomes:

$$L_{\mathrm{PODS}}(\theta, S) = \frac{1}{m} \sum_{i \in S} \frac{1}{|o_i|} \sum_{t=1}^{|o_i|}$$

$$\min \left[ \frac{\pi_\theta(o_{i,t} \mid p, o_{i,<t})}{\pi_{\theta_{\mathrm{fixed}}}(o_{i,t} \mid p, o_{i,<t})} \cdot a_{S,i}, \mathrm{clip} \left( \frac{\pi_\theta(o_{i,t} \mid p, o_{i,<t})}{\pi_{\theta_{\mathrm{fixed}}}(o_{i,t} \mid p, o_{i,<t})}, 1 - \varepsilon, 1 + \varepsilon \right) \cdot a_{S,i} \right].$$

Algorithm 1 outlines the PODS framework for GRPO with a single prompt $p$ in a training iteration. When training on a batch of multiple prompts, we simply apply the same procedure to each prompt and then concatenate the down-sampled rollouts and rewards. We conclude this section by presenting two trivial down-sampling strategies that can potentially be applied within PODS.

**Random down-sampling.** The rule $D_{\mathrm{rand}}$ uniformly selects $m$ indices from $\{1, 2, \ldots, n\}$ without replacement, thereby preserving the statistical properties of the original rollout distribution. In expectation, it yields the same parameter update as running standard GRPO on exactly $m$ rollouts.

---

**Algorithm 1** The PODS Framework for GRPO

---

**Input:** Models $\pi_\theta, \pi_{\theta_{\text{fixed}}}$, input prompt $p$, reward model $R$,
  Number of rollouts $n$, update size $m$, down-sampling rule $D$
  1: Independently sample $n$ rollouts $\mathbf{o} = (o_1, o_2, \ldots, o_n)$ using $\pi_{\theta_{\text{fixed}}}$ for prompt $p$
  2: Compute rewards $\mathbf{r} = (r_1, r_2, \ldots, r_n)$ using the reward model $R$
  3: Down-sample a set of $m$ rollouts $S \leftarrow D(\mathbf{o}, \mathbf{r}; m)$
  4: Update the policy using the GRPO-PODS objective $L_{\text{PODS}}(\theta, S)$
**Output:** An updated model $\pi_{\theta_{\text{updated}}}$

---

**Max-reward down-sampling.** The rule $D_{\text{maxr}}$ selects the $m$ rollouts with the highest rewards, concentrating on examples that exhibit the most desirable behavior. This should allow the model to learn primarily from successful reasoning patterns. However, as we show in Section 4, ignoring low-reward rollouts deprives the policy of negative feedback and can significantly degrade performance.

### 3.3 MAX-VARIANCE DOWN-SAMPLING

We now introduce *max-variance down-sampling*, a principled down-sampling rule that selects the most diverse and informative rollouts according to their reward distribution.

Specifically, $D_{\text{maxv}}$ chooses the subset $S$ of size $m$ that maximizes the empirical reward variance, i.e., $S = \arg\max_{|S|=m} \text{Var}(\{r_i \mid i \in S\})$. By spanning the full performance spectrum, it supplies strong contrastive signals between successful and unsuccessful reasoning paths. Recent work by Razin et al. (2025) provides an optimization-theoretic and empirical justification for this criterion.

A naive search would examine $O(\binom{n}{m})$ subsets. This is clearly infeasible for realistic $n$ and $m$. We prove, however, that the optimal subset can be found in $O(n \log n)$ time.

**Lemma 3.1.** *For a sorted list of rewards $r_1 \leq r_2 \leq \cdots \leq r_n$, the variance-maximizing subset of size $m$ always consists of the $k$ highest rewards and $(m - k)$ lowest rewards for some $k \in \{0, 1, \ldots, m\}$. That is,*

$$\text{Var}(\{r_1, \ldots, r_{m-k}\} \cup \{r_{n-k+1}, \ldots, r_n\}) = \max_{|S|=m} \text{Var}(\{r_i \mid i \in S\}).$$

**Proof of Lemma 3.1:** Let $S^* = \arg\max_{|S|=m} \text{Var}(\{r_i \mid i \in S\})$ be the optimal subset of size $m$. We will show that if $S^*$ is not of the form $\{1, \ldots, m - k\} \cup \{n - k + 1, \ldots, n\}$ for any $k$, then we can modify $S^*$ to obtain a new subset $S'$ of the same size with no smaller variance in rewards. By repeating this procedure, we can eventually reach a subset of this form.

Let $\mu$ be the mean of the rewards in $S^*$. Since the set $S^*$ does not take the form of $\{1, \ldots, m - k\} \cup \{n - k + 1, \ldots, n\}$ for any $k$, there exists either (i) an element $i \in S^*$ such that $i > 1, r_i \leq \mu$ and $i - 1 \notin S^*$, or (ii) an element $j \in S^*$ such that $j < n, r_j \geq \mu$ and $j + 1 \notin S^*$. That is, there exists an element in $S^*$, such that another element further from $\mu$ is not in $S^*$. We will show that we can swap them without decreasing variance.

For the ease of notation, we will denote $\text{Var}(\{r_i \mid i \in S\})$ as $\text{Var}(S)$ in this proof.

For case (i), let $S' = (S^* \setminus \{i\}) \cup \{i - 1\}$, and let $\mu'$ be the mean of the rewards in $S'$. Then

$$\begin{aligned} \text{Var}(S') - \text{Var}(S^*) &= \left(\frac{1}{m}\sum_{t \in S'} r_t^2 - \mu'^2\right) - \left(\frac{1}{m}\sum_{t \in S^*} r_t^2 - \mu^2\right) \\ &= \frac{1}{m}(r_{i-1}^2 - r_i^2) - (\mu'^2 - \mu^2) \\ &= \frac{1}{m}(r_{i-1} - r_i)(r_{i-1} + r_i) - (\mu' - \mu)(\mu' + \mu) \\ &= \frac{1}{m}(r_{i-1} - r_i)[(r_{i-1} + r_i) - (\mu' + \mu)] \geq 0. \end{aligned}$$

For case (ii), let $S' = (S^* \setminus \{j\}) \cup \{j + 1\}$, we can similarly show that $\text{Var}(S') - \text{Var}(S^*) \geq 0$.

In either case, we have shown that we can modify $S^*$ to obtain a new subset $S'$ of the same size that has no smaller variance in rewards. We can repeat this process until we reach a subset of the form $\{1, \ldots, m - k\} \cup \{n - k + 1, \ldots, n\}$ for some $k$. Thus, we conclude that there must exist one optimal subset of this form for some $k$. ∎

Lemma 3.1 naturally leads to a practical algorithm, Algorithm 2, for max-variance down-sampling. Moreover, it also offers intuition as to why maximizing variance is effective: the optimal subset contains the $k$ highest rewards and the $(m - k)$ lowest rewards, thereby capturing strong contrastive signals from both positive and negative examples.

---

**Algorithm 2** Max-Variance Down-Sampling

---

**Input:** Number of rollouts $n$, update size $m$, rollouts $\{o_1, o_2, \ldots, o_n\}$, rewards $\{r_1, r_2, \ldots, r_n\}$
1: Sort the rollouts by reward and get the sorted indices $ind \leftarrow \operatorname{argsort}(\{r_1, r_2, \ldots, r_n\})$
2: Let $S_{\text{ans}} \leftarrow \{ind_1, \ldots, ind_m\}$
3: **for** $k \in \{1, \ldots, m\}$ **do**
4:     Let $S_{\text{this}} \leftarrow \{ind_1, \ldots, ind_{m-k}\} \cup \{ind_{n-k+1}, \ldots, ind_n\}$
5:     Let $S_{\text{ans}} \leftarrow S_{\text{this}}$ **if** $\operatorname{Var}(\{r_i \mid i \in S_{\text{this}}\}) > \operatorname{Var}(\{r_i \mid i \in S_{\text{ans}}\})$
6: **end for**
**Output:** Selected indices $S_{\text{ans}}$ of rollouts

---

**Theorem 1.** *Algorithm 2 computes the max-variance down-sampling rule correctly. Moreover, it can be implemented in $O(n \log n)$ time.*

**Proof of Theorem 1:** The correctness of Algorithm 2 follows directly from Lemma 3.1.

For the time complexity, we first sort the rewards in $O(n \log n)$ time. To compute the variance of the selected rollouts, note that $\operatorname{Var}(\{x \mid x \in S_{\text{this}}\}) = \mathbf{E}_{x \in S_{\text{this}}}[x^2] - (\mathbf{E}_{x \in S_{\text{this}}}[x])^2$. We can maintain the prefix sums of the rewards and the squared rewards in $O(n)$ time. Then, for each $k$, we can compute the variance of the selected rollouts in $O(1)$ time using the prefix sums. Thus, the overall time complexity is $O(n \log n) + O(m) = O(n \log n)$. ∎

Theorem 1 shows that the max-variance down-sampling rule can be computed efficiently, which enables its practical application in GRPO-PODS. We conclude this section by noting an important special case of the max-variance down-sampling rule.

**Theorem 2.** *Let $m$ be an even integer. When the rewards are binary, selecting $m/2$ rollouts with the highest rewards and $m/2$ rollouts with the lowest rewards maximizes the variance of the rewards.*

**Proof of Theorem 2:** Let the number of rollouts with reward 1 be $k$. Then, the number of rollouts with reward 0 is $n - k$. If $k \leq m/2$, then any subset of $m$ rollouts contains at most $k$ rollouts with reward 1, and the variance is maximized by selecting these $k$ rollouts and any $(m - k)$ rollouts with reward 0. If $n - k \leq m/2$, then any subset of $m$ rollouts contains at most $(n - k)$ rollouts with reward 0, and the variance is maximized by selecting these $(n - k)$ rollouts and any $m - (n - k)$ rollouts with reward 1. Otherwise, any subset of $m/2$ rollouts with reward 1 and $m/2$ rollouts with reward 0 maximizes the variance. In all cases, we can select $m/2$ rollouts with the highest rewards and $m/2$ rollouts with the lowest rewards to maximize the variance. This concludes the proof. ∎

## 4 EXPERIMENTS

We evaluate PODS across diverse hardware configurations, model architectures, and model scales to demonstrate its generalizability and practical benefits. We test on two mathematical reasoning benchmarks—GSM8K (Cobbe et al., 2021) and MATH (Hendrycks et al., 2021)—using Qwen2.5 (Qwen et al., 2025) and Llama3.2 (MetaAI, 2024) models ranging from 3B to 7B parameters. Our experimental design covers both resource-constrained single-GPU setups and multi-GPU distributed training to validate PODS across different deployment scenarios. Table 1 describes our experimental configurations. To facilitate reproduction of our results, we have submitted our code to the supplementary materials and will release it upon publication of our work.

**Training infrastructure.** For settings (a-c), we use `Unsloth` (Daniel Han & team, 2023) with `TRL` (von Werra et al., 2020) for efficient LoRA (Hu et al., 2022) fine-tuning. For settings (d-e), we

| Setting | Benchmark | Model | Parameters | GPUs | Fine-tuning Method |
|---------|-----------|-------|------------|------|--------------------|
| (a) | GSM8K | Qwen2.5 | 3B | 1 L40S | LoRA (rank 64, $\alpha = 64$) |
| (b) | MATH | Qwen2.5 | 3B | 1 L40S | LoRA (rank 64, $\alpha = 64$) |
| (c) | GSM8K | Llama3.2 | 3B | 1 L40S | LoRA (rank 64, $\alpha = 64$) |
| (d) | GSM8K | Qwen2.5 | 3B | 8 H100s | Full-Parameter |
| (e) | GSM8K | Qwen2.5 | 7B | 8 A100s | Full-Parameter |

Table 1: Experimental configurations testing PODS across model scales, hardware constraints, and training paradigms. Settings (a-c) test resource-constrained scenarios with LoRA fine-tuning, while (d-e) evaluate full-parameter training with distributed setups.

implement distributed training with `DeepSpeed ZeRO-2` (Rajbhandari et al., 2020) and extend the `open-r1` library (HuggingFace, 2025) to support PODS on multiple devices.

**Rewards and evaluations.** We employ rule-based reward models that score rollouts, following standard practices in mathematical reasoning evaluation. Specifically, we reward an answer for correctness, format compliance, and the right number of thinking tags separately, resulting in a discrete but non-binary reward function. Details are provided in Appendix A.1.

**Section roadmap.** In Section 4.1, we compare the performance of GRPO and GRPO-PODS across five hardware and model settings listed in Table 1. We show that for all the settings we test, GRPO-PODS consistently outperforms GRPO in terms of performance as the training time increases. Then, in Section 4.2, we focus on setting (a), and analyze the effect of the rollout and update sizes $(n, m)$ on the performance of GRPO-PODS, providing empirical insights into how to choose the rollout and update sizes for GRPO-PODS. We present additional experiments about different down-sampling rules, and evaluation results about PODS's speed up ratio compared to GRPO and the average response length over the course of training in Appendices A.3 to A.5, respectively.

### 4.1 COMPARING GRPO-PODS TO BASELINE GRPO

We evaluate max-variance down-sampling PODS with GRPO against baseline GRPO using two experimental designs reflecting real-world constraints. For single-GPU settings (a-c), we compare against vanilla GRPO with matched training batch sizes $(m)$, where $m$ is selected to fit within memory. This corresponds to the comparison between GRPO and GRPO-PODS in Fig. 2. For distributed settings (d-e), we compare against GRPO with gradient accumulation (GRPO-GA)—the standard approach for scaling RLVR. In GRPO-GA, large batches are processed through multiple gradient accumulation steps, enabling updating on larger effective batch sizes at the cost of increased communication overhead and iteration time. We fix the total rollouts generated per prompt $(n)$ and compare GRPO-GA against GRPO-PODS. The detailed hyperparameters used for our experiments are listed in Appendix A.2. Results across all five configurations show consistent improvements.

Fig. 3 shows test accuracy over wall-clock training time across all configurations. PODS consistently achieves faster convergence: reaching the baselines' peak accuracies at least $1.7\times$ faster (see Appendix A.4 for complete results) while often exceeding final performance. These results demonstrate PODS's broad applicability across model scales (3B-7B), architectures (Qwen2.5, Llama3.2), and deployment scenarios, making it a practical improvement for RLVR systems using GRPO.

### 4.2 EFFECT OF ROLLOUT AND UPDATE SIZES $(n, m)$

A key practical question for PODS adoption is how to choose the rollout size $(n)$ and training batch size $(m)$. While a larger $n$ provides more diverse rollouts for selection, it also increases inference costs. Meanwhile, a smaller $m$ reduces update costs but may provide insufficient training signal. As shown in Fig. 4, we systematically study these trade-offs to provide deployment guidance.

Increasing rollout size $n$ exhibits diminishing returns with an optimal point around $n = 64$. Performance initially improves as larger pools enable better sample selection, but degrades beyond $n = 128$ due to two factors: (1) inference runtime grows significantly as GPU memory saturates, and (2) marginal improvements in rollout diversity plateau while computational overhead continues rising.

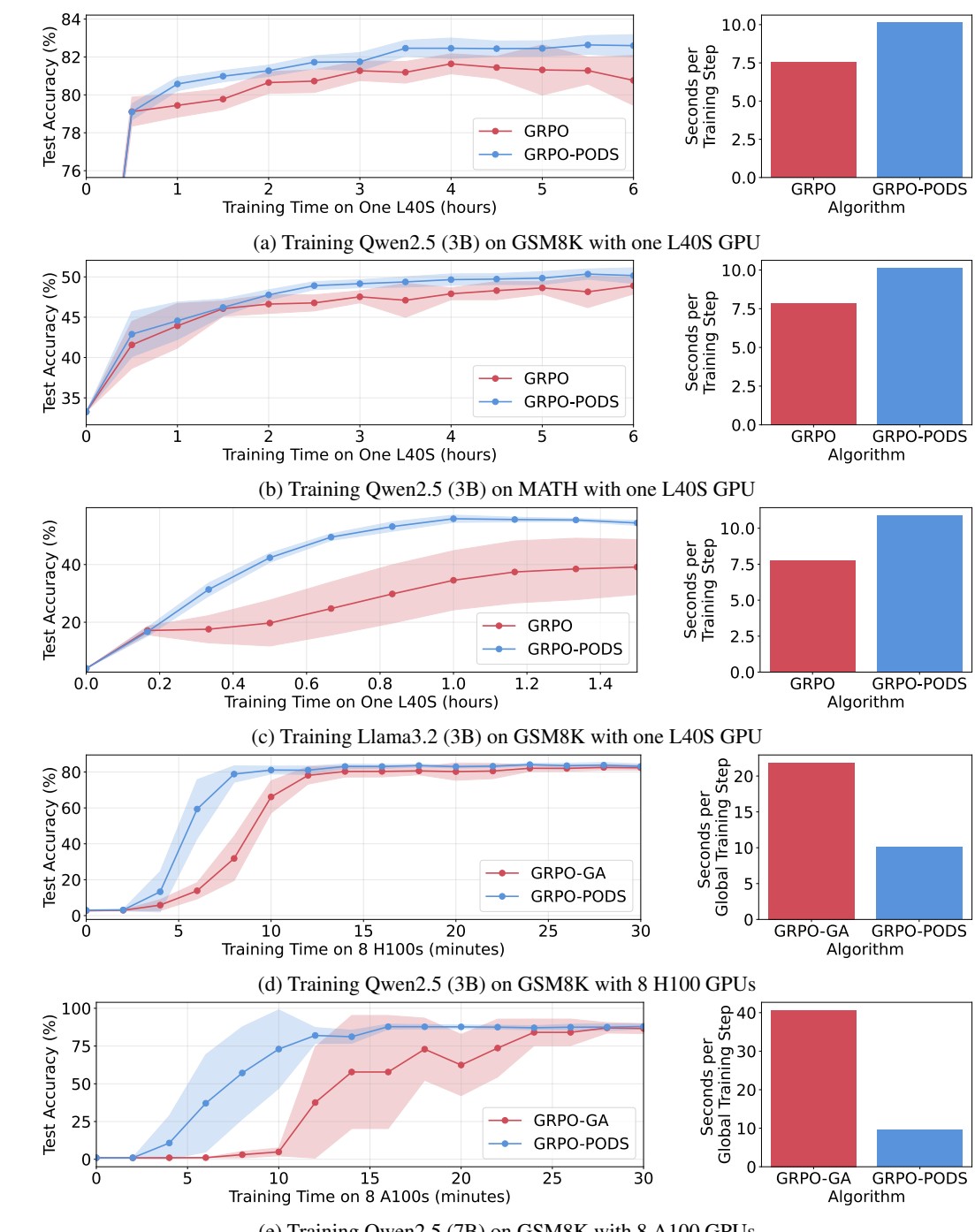

(a) Training Qwen2.5 (3B) on GSM8K with one L40S GPU

(b) Training Qwen2.5 (3B) on MATH with one L40S GPU

(c) Training Llama3.2 (3B) on GSM8K with one L40S GPU

(d) Training Qwen2.5 (3B) on GSM8K with 8 H100 GPUs

(e) Training Qwen2.5 (7B) on GSM8K with 8 A100 GPUs

Figure 3: Performance and per-step run time comparison of standard GRPO and GRPO-PODS with max-variance down-sampling across different datasets and hardware environments. For the performance comparison, the x-axis shows the training time, and the y-axis shows the accuracy on the test set. The shaded area represents 1.96 times the standard error of the mean.

Training batch size $m$ shows robust performance across a wide range, with minimal degradation until very small values ($m \leq 4$). This suggests PODS' max-variance selection maintains effective learning signals even with aggressive down-sampling ratios up to 16 where $n = 64, m = 4$.

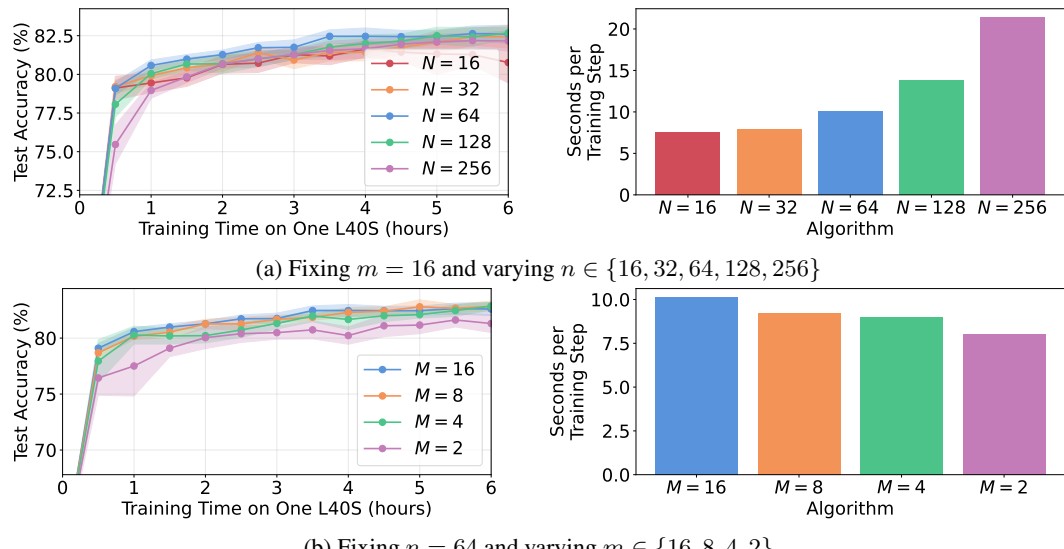

(a) Fixing $m = 16$ and varying $n \in \{16, 32, 64, 128, 256\}$

(b) Fixing $n = 64$ and varying $m \in \{16, 8, 4, 2\}$

Figure 4: Performance and per-step run time comparison of GRPO-PODS with max-variance down-sampling across different settings of $n$ and $m$. The training is conducted on the GSM8K dataset with one L40S. For the performance comparison, the x-axis shows the training time, and the y-axis shows the accuracy on the test set. The shaded area represents 1.96 times the standard error of the mean.

**Practical guidelines.** These results suggest down-sampling ratio of 2 to 4 (i.e., $m = 16$ and $n = 32$ or 64 in Fig. 4a) provides an effective balance of performance and efficiency. For resource-constrained settings, aggressive down-sampling ratios up to 16 remain viable, while memory-rich environments can benefit from larger rollout pools up to hardware limits.

## 5 CONCLUSION AND DISCUSSION

We introduced **PODS**—a lightweight, algorithm-agnostic framework that addresses a fundamental bottleneck in modern RLVR training: the asymmetry between embarrassingly parallel rollout generation and memory-intensive policy updates. PODS generates large batches of rollouts in parallel and updates the policy on only an informative subset chosen by the max-variance rule. Our analysis shows that the optimal subset can be found in $O(n \log n)$ time. This simple yet principled approach consistently outperforms standard GRPO under equal wall-clock budgets, delivers at least a $1.7\times$ speedup and reaching higher final accuracy across diverse model architectures, scales, and deployment scenarios. Our ablation study shows that the performance of PODS is robust over a wide range of down-sampling ratios provided $m$ is not too small, empirically confirming our method's efficacy.

**Limitations.** Our evaluation focuses on mathematical-reasoning tasks with verifiable rewards. Other domains such as open-ended dialogue or code generation may exhibit distinct dynamics of the algorithms. Moreover, in workloads that demand greater prompt diversity, similar gains might be obtained by processing more prompts per iteration with fewer rollouts per prompt and accumulating gradients across prompts—an alternative path to address the inference-update asymmetry. Finally, because PODS alters the training rollout distribution through selective down-sampling, it behaves off-policy and may be unsuitable when strict on-policy guarantees are required.

**Future work.** The algorithm-agnostic nature of PODS enables integration with value-based methods like PPO and emerging RL approaches. Exploring whether PODS can enhance state-of-the-art model performance represents a promising research problem. Additionally, investigating adaptive down-sampling strategies that evolve throughout training could further optimize the learning dynamics. Exploring theoretically principled approaches to balance the trade-off between prompt diversity and rollout quantity per prompt also warrants investigation.

**Ethics statement.** We anticipate our work will primarily have positive social impact by improving the computational efficiency and effectiveness of RL training for LLMs, potentially democratizing access to high-quality reasoning models. However, by lowering the computational barriers to training powerful reasoning systems, our method may accelerate capabilities that could be misused. This heightens the importance of responsible release practices to mitigate harmful behaviors. Our open-source release of code and experimental frameworks aims to facilitate reproducibility while encouraging informed and safe adoption within the research community.

**Reproducibility statement.** We list the key hyperparameters in Appendix A.2, describe the rewards used in Appendix A.1, and an anonymized version of the code used to run the experiments in this paper is attached to our submission (as supplementary material) on OpenReview. We will publicly release the code on GitHub, and include a link to the repo in the next version of our paper. We note that all of the datasets used in this paper are open-source, and the models we use are all open-weight and available publicly on HuggingFace.

**LLM usage statement.** In this work, we used LLMs as an assist tool in polishing the language of our writing in the paper and auto-completing some of our evaluation code.

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

# A  ADDITIONAL EXPERIMENTAL DETAILS

## A.1  REWARD FUNCTIONS

The specific reward functions we use in our experiments are listed below.

**Accuracy (1.0 for correct, 0.0 for incorrect):** Mathematical correctness using LaTeX parsing and symbolic verification. The reward function extracts mathematical expressions from both the model's response and ground truth solution, then uses symbolic verification to determine equivalence.

**Format (1.0 for compliant, 0.0 for non-compliant):** Structured response formatting that requires reasoning to be enclosed within `<think>` tags and the final answer within `<answer>` tags, following the exact pattern `<think>\n...\n</think>\n<answer>\n...\n</answer>`.

**Tag count (0.0 to 1.0 partial credit):** Formatting rewards for proper XML tag usage. The model receives 0.25 points each for correct placement of `<think>\n`, `\n</think>\n`, `\n<answer>\n`, and `\n</answer>` tags, allowing partial credit for partially correct formatting.

## A.2  HYPERPARAMETERS

In Table 2, we list the key hyperparameters we use for different experimental settings.

Table 2: Hyperparameters for different experimental settings.

| Setting | (a) | (b) | (c) | (d) | (e) |
|---|---|---|---|---|---|
| Optimizer | AdamW | AdamW | AdamW | AdamW | AdamW |
| Max Sequence Length | 1024 | 1024 | 1024 | 2048 | 2048 |
| Lora Rank | 64 | 64 | 64 | N/A | N/A |
| Lora Alpha | 64 | 64 | 64 | N/A | N/A |
| KL Coefficient | 0.00 | 0.00 | 0.04 | 0.00 | 0.00 |
| Learning Rate | $5 \times 10^{-6}$ | $5 \times 10^{-6}$ | $2 \times 10^{-6}$ | $2 \times 10^{-5}$ | $1.5 \times 10^{-5}$ |
| Weight Decay | 0.1 | 0.1 | 0.1 | 0.1 | 0.1 |
| Grad Clipping | 1.0 | 1.0 | 1.0 | 1.0 | 1.0 |
| GA Steps (GRPO-PODS) | 1 | 1 | 1 | 4 | 4 |
| Rollout Batch Size (GRPO-PODS) | 64 | 32 | 64 | 128 | 128 |
| Update Batch Size (GRPO-PODS) | 16 | 8 | 16 | 32 | 32 |
| Effective $n$ (GRPO-PODS) | 64 | 32 | 64 | 512 | 512 |
| Effective $m$ (GRPO-PODS) | 16 | 8 | 16 | 128 | 128 |
| Down-Sampling Ratio | 4 | 4 | 4 | 4 | 4 |
| GA Steps (GRPO) | 1 | 1 | 1 | N/A | N/A |
| Rollout Batch Size (GRPO) | 16 | 8 | 16 | N/A | N/A |
| Update Batch Size (GRPO) | 16 | 8 | 16 | N/A | N/A |
| Effective $n$ (GRPO) | 16 | 8 | 16 | N/A | N/A |
| Effective $m$ (GRPO) | 16 | 8 | 16 | N/A | N/A |
| GA Steps (GRPO-GA) | N/A | N/A | N/A | 16 | 16 |
| Rollout Batch Size (GRPO-GA) | N/A | N/A | N/A | 32 | 32 |
| Update Batch Size (GRPO-GA) | N/A | N/A | N/A | 32 | 32 |
| Effective $n$ (GRPO-GA) | N/A | N/A | N/A | 512 | 512 |
| Effective $m$ (GRPO-GA) | N/A | N/A | N/A | 512 | 512 |

**Note on gradient accumulation.** For experiment settings **(d)** and **(e)**, we ensured a fair comparison between GRPO-PODS and GRPO-GA by matching the total number of rollouts (effective $n$) generated per prompt. This was done by equating the product of rollout batch size and GA steps across both methods. In the `open-r1` implementation, GA steps determine both rollout generation and training updates. For example, with rollout batch size 128 and GA steps 4, the effective $n$ is $128 \times 4 = 512$. We fixed this number at 512 for both GRPO-PODS and GRPO-GA.

For GRPO-PODS, each rollout batch was down-sampled by a factor of $4$, resulting in an update batch size of 32 and an effective $m$ of $32 \times 4 = 128$. Because down-sampling is applied directly after generating each batch rather than after aggregation, GRPO-GA must increase GA steps by $4\times$ (from $4$ to 16) to maintain the same effective $n$. This adjustment ensures that both variants process an equal number of rollouts while respecting their structural differences.

## A.3 COMPARING DIFFERENT DOWN-SAMPLING RULES

We study the effect of different down-sampling rules on the performance of GRPO-PODS in this section. We conduct experiments on the GSM8K dataset with one L40S GPU. We set the rollout size $n = 64$ and the update size $m = 16$, and we compare three different down-sampling rules: (1) max-variance down-sampling, (2) max-reward down-sampling, and (3) random down-sampling. The results are shown in Fig. 5. We observe that the max-variance down-sampling rule consistently outperforms both the max-reward and random down-sampling rules across all settings. This indicates that the max-variance down-sampling rule is effective in selecting informative rollouts for training.

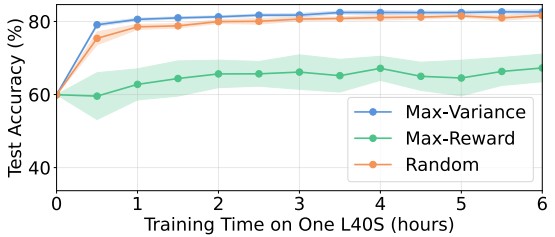 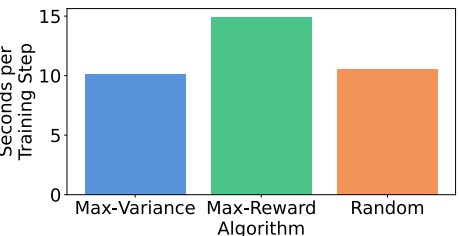

Figure 5: Performance and per-step run time comparison of GRPO-PODS with the max-variance, max-reward and random down-sampling rules. The training is conducted on the GSM8K dataset with one L40S. For the performance comparison, the x-axis shows the training time, and the y-axis shows the accuracy on the test set. The shaded area represents 1.96 times the standard error of the mean.

## A.4 PODS' SPEED UP RATIO OVER GRPO

In Fig. 3, we observe that GRPO-PODS consistently outperforms GRPO in terms of performance as the training proceeds. For each of the five plots in Fig. 3, we compute the speed up ratio of GRPO-PODS over GRPO, i.e., the ratio between the time taken by GRPO and that taken by GRPO-PODS to reach $0.99\times$ the peak performance of GRPO. The results are shown in Table 3. We observe that our method achieves a speed up ratio between $1.7\times$ and $3.0\times$ over GRPO across the settings.

Table 3: Speed up ratio of GRPO-PODS over GRPO in Fig. 3.

| Setting | (a) | (b) | (c) | (d) | (e) |
|---|---|---|---|---|---|
| Speed Up Ratio | $2.0\times$ | $2.0\times$ | $3.0\times$ | $1.7\times$ | $1.7\times$ |

## A.5 AVERAGE COMPLETION LENGTH OVER TIME

We include additional evaluation of the average completion length over the training time for each of the experiments we conduct in Section 4. We present the average completion length results in Figs. 6 to 8, in correspondence to Figs. 3, 4 and 5 respectively. In most of the cases, we observe that the average completion length stays relatively stable over the training time.

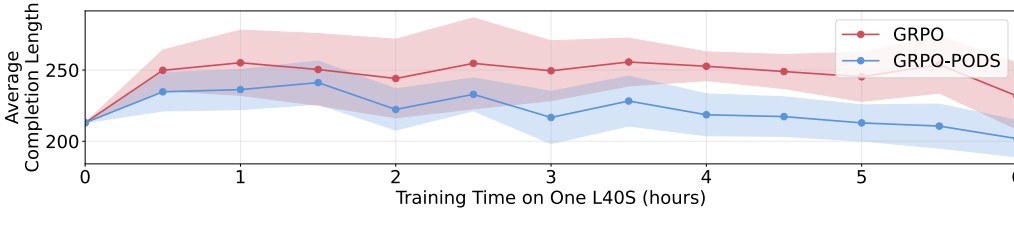

(a) Training Qwen2.5 (3B) on GSM8K with one L40S GPU

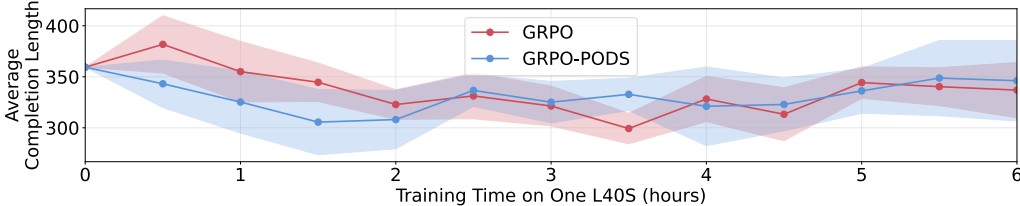

(b) Training Qwen2.5 (3B) on MATH with one L40S GPU

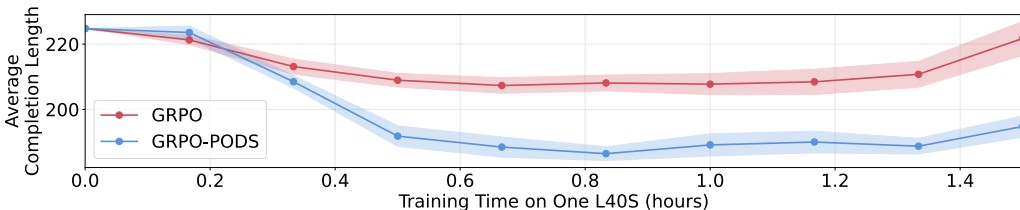

(c) Training Llama3.2 (3B) on GSM8K with one L40S GPU

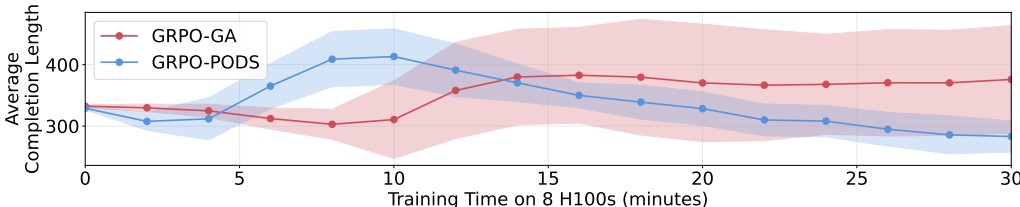

(d) Training Qwen2.5 (3B) on GSM8K with 8 H100 GPUs

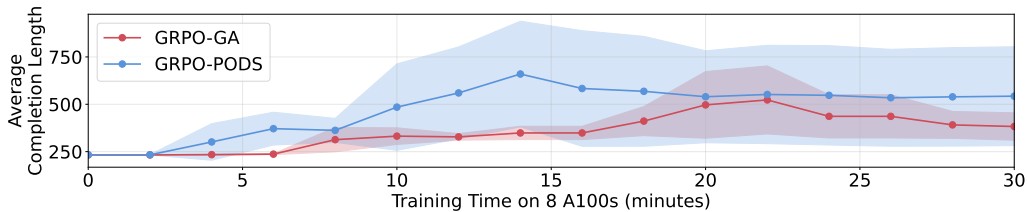

(e) Training Qwen2.5 (7B) on GSM8K with 8 A100 GPUs

Figure 6: Average completion length over time of the trained models in Section 4.1's experiments. The x-axis shows the training time, and the y-axis shows the average completion length in tokens. The shaded area represents 1.96 times the standard error of the mean.

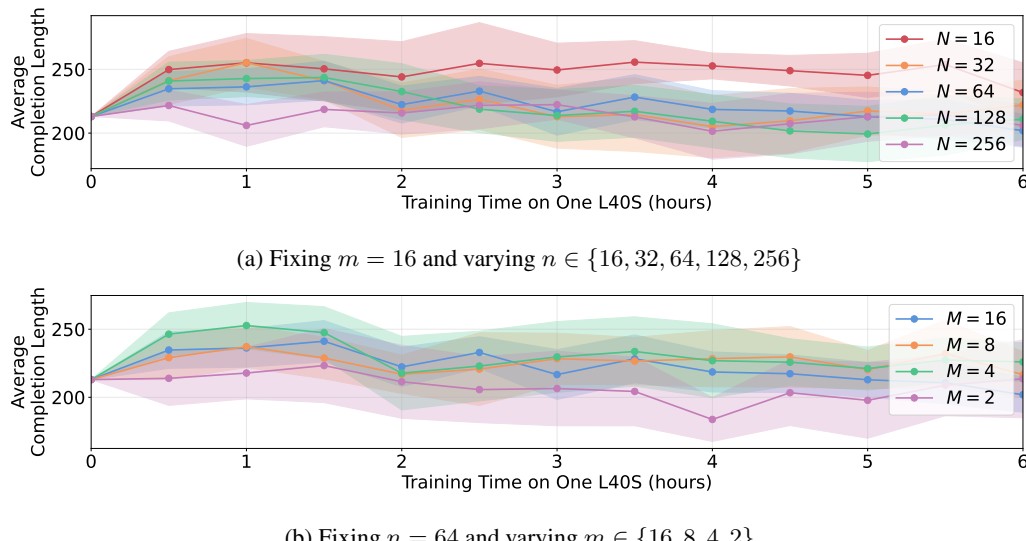

(a) Fixing $m = 16$ and varying $n \in \{16, 32, 64, 128, 256\}$

(b) Fixing $n = 64$ and varying $m \in \{16, 8, 4, 2\}$

Figure 7: Average completion length over time of the trained models in Section 4.2's experiments. The x-axis shows the training time, and the y-axis shows the average completion length in tokens. The shaded area represents 1.96 times the standard error of the mean.

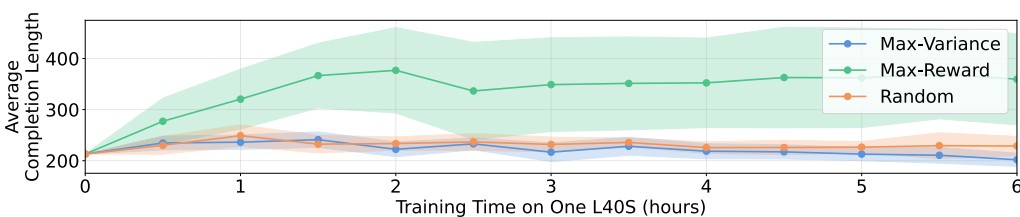

Figure 8: Average completion length over time of the trained models in Appendix A.3's experiments. The x-axis shows the training time, and the y-axis shows the average completion length in tokens. The shaded area represents 1.96 times the standard error of the mean.

