# OpenReview forum: "Not All Rollouts are Useful: Down-Sampling Rollouts in LLM Reinforcement Learning"
_ICLR.cc/2026/Conference — ICLR 2026 Conference Withdrawn Submission_

### Official Review · Reviewer_ddiB · 2025-10-19

**Soundness:** 2
**Presentation:** 2
**Contribution:** 1
**Rating:** 2
**Confidence:** 4

**Summary:**

This paper introduces PODS, a data selection framework designed to improve the efficiency of GRPO training. The key idea is that rollout generation can be efficiently parallelized via batching, whereas policy updates become costly with large rollout sizes. To address this, PODS downsamples generated rollouts by selecting those with the lowest and highest rewards, thereby maximizing reward variance within each training batch.

**Strengths:**

+ GRPO efficiency is an important and timely problem in RLHF.
+ The paper is overall well-written and well-structured.
+ Empirical results show encouraging performance improvements.

**Weaknesses:**

- The contribution is relatively incremental, both in the data selection strategy and in algorithmic design.
- The evaluation is limited in scope: benchmarks focus only on GSM8K and MATH, and baselines are restricted to the naive GRPO implementation. Including code generation or reasoning benchmarks would strengthen the results. Also, comparative analysis could be more comprehensive using relevant baselines, such as GRESO [1].

**Questions:**

Thank you for submitting this work. The paper tackles an important problem in RLHF training efficiency, but I am concerned that the novelty is limited and that key design choices are not sufficiently justified. Below are specific comments and questions:

- Q1: he complexity of Algorithm 1 should be O(mlogn), since maintaining the m lowest and highest rollouts in a priority queue avoids sorting the entire list. Could the authors clarify?

- Q2: How does PODS perform on more diverse tasks and benchmarks, such as code generation or reasoning datasets? Including stronger baselines (e.g., GRESO [1]) would make the results more convincing.

- Q3: Could the authors elaborate on why PODS leads to better final model performance? This improvement and claim seem highly sensitive to the choice of the hyperparameter m.

Reference:

[1] Act Only When It Pays: Efficient Reinforcement Learning for LLM Reasoning via Selective Rollouts, NeurIPS 2025 / arXiv:2506.02177.

---

> ### Author Response · Authors · 2025-11-19
> **Author Rebuttal to Review ddiB**
>
> We thank Reviewer ddiB for their feedback and for acknowledging the importance of GRPO efficiency. We would like to address the reviewer's specific questions and comments on novelty.
>
> ### **1\. On the Complexity of Algorithm 1/2 (Q1):**
>
> We thank the reviewer for the comment on complexity. We believe there are two misunderstandings here.
>
> First, the complexity of using a priority queue (or heap) to find the $m$ lowest and highest elements from $n$ items would be $O(n \\log m)$, not $O(m \\log n)$ as the reviewer suggests.
>
> Second, and more importantly, our algorithm does *not* simply select the $m/2$ lowest and $m/2$ highest rollouts. A priority-queue-based approach ($O(n \\log m)$) would only be sufficient if that were the case (which is only guaranteed for binary rewards, Thm 2). As shown in **Lemma 3.1**, the true max-variance subset is the $k$ highest and $(m-k)$ lowest *for some* $k \\in \\{0, ..., m\\}$.
>
> To find this optimal $k$, our $O(n \\log n)$ algorithm first sorts all $n$ rollouts. It then correctly solves the problem by checking all $m+1$ possible splits, which takes $O(m)$ time using prefix sums (as detailed in the Thm 1 proof). A priority-queue approach cannot efficiently test all these $m$ splits.
>
> While other selection algorithms exist (e.g., expected $O(n)$), our central goal was to ensure the down-sampling step is **polynomial** and not an exponential bottleneck. Given that the $O(n \\log n)$ time of our simple, sort-based algorithm is **already negligible** in practice compared to the high cost of inference and policy updates (as shown in Fig 1), we believe simplicity and correctness outweigh the marginal gains of further optimizing this non-bottleneck step.
>
> ### **2\. On Incremental Contribution (W1):**
>
> We respectfully disagree that the contribution is incremental. While the *components* (over-generation, down-sampling) are simple, the core contribution is identifying and solving the *systems-level asymmetry* between parallel-friendly inference and memory/communication-bound policy updates.
>
> The novelty lies in: a) Identifying this specific asymmetry as the key bottleneck for RLVR wall-clock time (Fig 1). b) Proposing a *principled* (max-variance) and *efficient* ($O(n \\log n)$) selection rule that demonstrably outperforms naive (random, max-reward) selection. This rule is *guaranteed* to capture rare, high-signal events, unlike naive sampling. c) Demonstrating that this system-aware approach leads to significant (1.7x-3.0x) wall-clock speedups, not just minor accuracy gains.
>
> ### **3\. On Missing Baselines (GRESO) and Scope (Q2):**
>
> We thank the reviewer for pointing out the highly relevant work \[1\] GRESO (NeurIPS 2025\) and will include this work in our references. However, if the reviewer reads the GRESO paper carefully, they will note that GRESO **cites our work** (their Section 2), as our paper has been available as a preprint. This validates that our work is a foundational contribution to this line of inquiry.
>
> On the scope, we agree that expanding to other benchmarks like code generation is an important next step. However, we would like to clarify that our current tasks are not as simple as they may appear. The reward function (Appendix A.1) is *not* purely binary; it is a discrete, non-binary function that includes partial credit for formatting. The success of our method on this more complex signal suggests it is not limited to simple 0/1 tasks.
>
> ### **4\. On Better Final Performance (Q3):**
>
> This is an excellent question. We hypothesize that PODS achieves better final performance because the max-variance selection rule provides a *cleaner, higher-signal gradient* per update.
>
> To be concrete: in a highly unbalanced batch (e.g., 1 failure and 63 successes), max-variance **guarantees** the selection of that rare failure, providing a critical learning signal that random sampling would almost certainly miss. By preventing this kind of signal loss and filtering out redundant "average" rollouts, PODS provides a more robust and effective gradient at each step. This is supported by our ablation (Fig. 5), which shows max-variance strongly outperforms random sampling.
>
> This finding—that training on a strategically selected high-variance subset yields superior final performance compared to training on all available rollouts—shows that the benefit isn't merely computational efficiency, but **improved learning quality**. This aligns with established RL literature on advantage-based filtering \[2\].
>
> ### **References**
>
> \[1\] Act Only When It Pays: Efficient Reinforcement Learning for LLM Reasoning via Selective Rollouts, NeurIPS 2025 / arXiv:2506.02177.
>
> \[2\] Cusumano-Towner, Marco, et al. "Robust autonomy emerges from self-play." arXiv preprint arXiv:2502.03349 (2025).

---

### Official Review · Reviewer_K7yh · 2025-10-26

**Soundness:** 3
**Presentation:** 3
**Contribution:** 3
**Rating:** 6
**Confidence:** 1

**Summary:**

This work introduces PODS (Parallel Optimized Down-Sampling), a lightweight and algorithm-agnostic framework designed to tackle a core inefficiency in modern Reinforcement Learning with Verifiable Rewards (RLVR): the mismatch between highly parallelizable rollout generation and memory-constrained, sequential policy updates. PODS addresses this by generating large batches of rollouts in parallel and selectively updating the policy on a small, informative subset chosen via a max-variance selection rule.

Despite its simplicity, PODS delivers consistent improvements: under identical wall-clock time budgets, it outperforms standard GRPO, achieves at least a 1.7× training speedup, and attains higher final accuracy across diverse model architectures, scales, and deployment settings.

**Strengths:**

(1) Well-written
(2) Detailed experiment
(3) The problems related to training efficiency that have been solved are distinctive and seem valuable to the industrial sector

**Weaknesses:**

N/A

**Questions:**

I do not know this specialized research field very well. I will adjust my score and optimize my review document based on the evaluations of other expert reviewers and my performance during the rebuttal period.

---

> ### Author Response · Authors · 2025-11-19
> **Author Rebuttal to Review K7yh**
>
> We thank Reviewer K7yh for their positive feedback. We are glad they found the paper well-written and the problem we solve to be distinctive and valuable. We appreciate their time and assessment.

---

### Official Review · Reviewer_XkuY · 2025-10-30

**Soundness:** 3
**Presentation:** 3
**Contribution:** 2
**Rating:** 2
**Confidence:** 3

**Summary:**

This paper aims at the computational inefficency RLVR for large languange models. This paper identify a key asymmetry between the paralleizable rollout generation phase and memory-intensive policy update phase. To mitigate this, this paper proposes PODS, a general framework that generates rollouts but selective update the policy with a subset of the rollouts. Meanwhile, thie paper introduces a max-variance down-sampling method to select rollout with maximum reward diversity. Experiments with GRPO on GSM8k and MATH shows that PODS maintains comparable accuracy performance and more than 1.7x faster convergence.

**Strengths:**

1. Originality: This paper tackles an underexplored issue in LLM RL. The proposed PODS framework and max-variance down-sampling criterion represent an interesting aspect of efficiency optimization. This work distincts from existing prompt-selection or gradient-accumulation approaches. The idea of selective subset section is conceptually simple and original.
2. Quality: The methodology is clearly formalized with simple mathematical justification. Theoretical analysis provides clarity and computational guarantees for the proposed down-sampling rule. The experiments are comprehensive, spanning different model sizes, datasets, and hardware setups.
3. Clarity: The paper is well-written and easy to follow. The motivation for the problem is clearly illustrated. The presentation of algorithms and visual comparisons are well-structured.
4. Significance: The work addresses a bottleneck in scaling reinforcement learning for LLMs, reducing memory and communication overhead during policy updates. Given the rising computational costs in reasoning-focused RL for LLMs, the proposed method is timely and practically impactful. This framework can be integrated with other RL variants, increasing its potential influence and relevance for large-scale LLM training.

**Weaknesses:**

1. LImited scope empirical validation. As mentioned in the limitation section, the evaluation is conducted on mathematical reasoning tasks (GSM8K, MATH) with rule-based reward models. These tasks have well-defined, verifiable rewards, which may overstate the benefits of the method. The paper would be significantly strengthened by including results on other multiple tasks where reward distributions are noisier and less binary.
2. Dependence on single baseline RLVR algorithm. Although the method is claimed to be algorithm-agnostic, all experiments and analyses are built around GRPO. It remains unclear whether the same variance-based selection criterion would hold for other RL frameworks.  Demonstrating adaptability across multiple RL algorithms would make the contribution more convincing.
3. Lack of deeper theoretical justification of learning dynamics. The paper provides a sound combinatorial analysis for max-variance selection but does not connect this formally to expected policy improvement or gradient variance reduction. Without a theoretical link between rollout diversity and learning efficiency, the variance criterion, while intuitive, remains heuristic.
4. Limited discussion of potential trade-offs: While the paper reports 1.7×–3× speedups, it provides little detail about wall-clock breakdowns (e.g., inference vs. update time) or the communication cost savings. Furthermore, potential drawbacks such as off-policy bias or degradation of gradient fidelity with extreme down-sampling ratios are acknowledged but not empirically investigated.
5. Meanwhile, the LLM used in this paper is simply 3B and 7B, of which the inference time consumption and training time is fair. However for larger LLMs, the inference time and training time increase significantly. The time complexity of subset selection, nlogn, is not that important, especially for rollout number of 64 and subset size 16. The reduction of average training seconds per training step is mainly caused by the reduction of training data. Therefore, the time complexity of this subset selection is fair theoretically but limited in actual training.

**Questions:**

1. The evaluation focuses exclusively on GSM8K and MATH which have verifiable and relatively noise-free rewards. Can the authors discuss any preliminary evidence or theoretical reasoning suggesting that the max-variance down-sampling rule remains effective under noisy or sparse reward distributions?
2. While PODS is described as algorithm-agnostic, could the authors explain what modifications, if any, would be required to adapt PODS to reinforce++ or to reinforcement learning from human feedback (RLHF) setups? Please provide one of the demonstration results.
3. Since PODS alters the effective training distribution by selective sampling, does this introduce off-policy bias relative to standard on-policy GRPO?
4. Have the authors considered or tested any correction mechanisms (e.g., importance weighting) to mitigate this potential bias? A short theoretical justification or empirical comparison could help assess the trade-off between efficiency and policy fidelity.
5. The proposed max-variance rule is intuitive and empirically effective, but the paper does not formally connect it to expected policy improvement. Could the authors provide analytical or empirical arguments showing that higher reward variance indeed correlates with more informative gradients or faster convergence in GRPO-like updates?
6. The paper reports wall-clock speedups, but it would be helpful to see a breakdown of inference time, policy-update time, subset data selection time, especially in distributed setups.
7. How does max-variance down-sampling compare to other principled criteria such as entropy-based selection, uncertainty sampling, or advantage-based weighting?

---

> ### Author Response · Authors · 2025-11-19
> **Author Rebuttal to Review XkuY**
>
> We thank Reviewer XkuY for their feedback and for acknowledging the originality, clarity, and practical significance of our work in addressing the RLVR computational bottleneck.
>
> We would like to address several points, in particular a core misunderstanding regarding the *source* of our reported speedup, and provide answers to the reviewer's specific questions.
>
> ### **W1 & Q1 (Limited Scope & Noisy Rewards):**
>
> We would like to clarify two points. First, our focus is deliberately on the **Reinforcement Learning with Verifiable Rewards (RLVR)** setting, as it has its own distinct computational challenges that we aim to solve.
>
> Second, our reward function is **not** noise-free or purely binary. As we detail in **Appendix A.1**, the reward is a discrete, non-binary function that includes partial credit for formatting (e.g., correct use of tags). This multi-part reward acts as a more complex signal than a simple 0/1 correctness. The fact that max-variance selection works so well on this more complex signal is strong "preliminary evidence" (Q1) that it is robust and not limited to simple binary signals.
>
> ### **W2 & Q2 (Single Baseline Algorithm & Adaptation to RLHF):**
>
> This is a fair question. We chose GRPO because it is a (or *the*) foundational and widely-used algorithm in the **RLVR** setting, which is the explicit focus of our paper. Adapting our *framework* to RLHF is an interesting direction, but it is a different problem domain than the RLVR setting we address.
>
> ### **W3 & Q5 (Lack of Deeper Theoretical Justification):**
>
> The reviewer asks for a formal link between max-variance and *gradient* variance or policy improvement. While a deep theoretical analysis of the gradient is a valuable but distinct research question, our criterion is not merely "heuristic."
>
> It has a *principled* combinatorial justification (Lemma 3.1) and, more importantly, a practical one (Q5). By maximizing reward variance, our method **guarantees** the selection of rare, high-signal events. For example, in a batch of 64 rollouts with 1 failure and 63 successes, max-variance *will* select that single failure, providing a critical learning signal. Random sampling, by contrast, would miss this signal \~98% of the time (with m=16). This direct preservation of high-signal, low-probability events is a clear, non-heuristic justification for *why* our rule leads to more informative gradients and faster convergence.
>
> ### **W4 & Q6 (Breakdown of Runtimes):**
>
> We must respectfully disagree with the claim that there is a "Limited discussion of... wall-clock breakdowns" or a lack of detail.
>
> The *entire motivation* for our paper is based on this breakdown, which is provided in detail in **Figure 1**. That figure explicitly plots "Wall Clock Time (s)" for "Inference" vs. "Policy Update (Underutilized)" vs. "Policy Update (Gradient Accumulation)". This figure is the central empirical evidence for the computational asymmetry that our entire method is designed to solve. We refer the reviewer to this figure for the exact breakdown they are requesting.
>
> ### **W5 (Source of Speedup and Complexity):**
>
> We thank the reviewer for this point, and we are in agreement. The reviewer is correct that for large models, the $O(n \\log n)$ selection time is negligible compared to the inference and update costs. This is a key *feature* of our method: the selection overhead is minimal, which is precisely why it is practical. Our contribution is an $O(n \\log n)$ algorithm, and we *wanted* this to be negligible, as an exponential-time selection would be a practical bottleneck.
>
> The reviewer is also correct that the reduction in per-step time comes from reducing the update data (using a smaller $m$). Our hypothesis is that beyond a certain point, more rollouts provide diminishing returns ("marginal signal"). By using a principled rule (max-variance) to select a small, high-signal subset $m$, we can *reduce the training data* for the update phase, save time, and move to the next training batch faster, all while *improving* (or maintaining) final model accuracy.

---

> > ### Author Response · Authors · 2025-11-19
> > **Author Rebuttal to Review XkuY Continued**
> >
> > ### **Q3 & Q4 (Off-Policy Bias):**
> >
> > This is a correct technical observation. However, this is a principled and motivated choice, not an accidental one. In practice, empirical performance often outweighs strict adherence to theoretical properties, especially when the deviation is well-motivated. Recent work has, in fact, re-evaluated GRPO itself as having a "native off-policy interpretation" \[2\]. Our strong empirical results (Fig. 3)—where PODS not only converges faster but also matches or *improves* upon the final accuracy of the baseline—validate this choice, aligning with other recent work showing the benefits of off-policy GRPO \[1\]. This demonstrates that the practical gains from a better learning signal and superior system efficiency far outweigh any theoretical concerns about off-policyness in this context.
> >
> > ### **Q7 (Comparison to other criteria):**
> >
> > These are excellent suggestions. Criteria like entropy-based selection or uncertainty sampling can be seen as different options *within* the general PODS framework. For a specific application, one may have motivation to use a different rule. Our paper proposed and validated max-variance as a simple, fast, and effective rule for this domain. We also provide a direct comparison to two other key baselines—random sampling and max-reward sampling—in **Appendix A.3 (Figure 5\)**, showing that max-variance is empirically superior.
> >
> > ### **References**
> >
> > \[1\] Mroueh, Y., et al. "REVISITING GROUP RELATIVE POLICY OPTIMIZATION: INSIGHTS INTO ON-POLICY AND OFF-POLICY TRAINING." arXiv preprint arXiv:2505.22257 (2025).
> >
> > \[2\] Yao, C., et al. "Group-Relative REINFORCE Is Secretly an Off-Policy Algorithm: Demystifying Some Myths About GRPO and Its Friends." arXiv preprint arXiv:2509.24203 (2025).

---

### Official Review · Reviewer_ByiU · 2025-11-01

**Soundness:** 3
**Presentation:** 3
**Contribution:** 3
**Rating:** 6
**Confidence:** 4

**Summary:**

This paper targets a very concrete bottleneck in reinforcement learning for LLM reasoning with verifiable rewards: rollout generation is cheap and highly parallelizable, but policy updates (GRPO/PPO-style) are memory- and communication-bound, so we can’t just “generate more rollouts” to improve sample quality. The authors propose PODS (Policy Optimization with Down-Sampling): for each prompt, generate a large pool of rollouts, score them with the verifiable reward, and then select only a small, most informative subset for the policy update. The key technical piece is a max-variance selection rule: choose the size-𝑚 subset whose rewards have the largest variance, which they show always corresponds to “take some of the lowest and some of the highest” rollouts; this yields an efficient $O(nlogn)$ algorithm. Plugged into GRPO, PODS achieves 1.7×–3× faster wall-clock to the same or better accuracy on GSM8K/MATH and across Qwen2.5 and Llama 3.2 models, on both single-GPU (LoRA) and multi-GPU (full-param) setups.

**Strengths:**

1. Very well-motivated problem. The paper identifies a real training-systems bottleneck: inference scales, but updates don’t.
2. Simple, plug-in idea. PODS is architecturally lightweight: keep your GRPO pipeline, just over-generate and then down-sample. This makes adoption easy.
3. Principled selection rule. Instead of just “pick top-k,” they argue that we want reward diversity. The max-variance formulation + structure theorem is a nice, clean piece of theory that justifies the heuristic.
4. Strong empirical evidence. Multiple models, multiple hardware regimes , and realistic tasks. The improvement is in wall-clock, not just final accuracy, which matters in practice.
5. Clear ablations. They study both the number of generated rollouts n and the number of kept rollouts m, and compare to random and max-reward selection. The proposed max-variance rule consistently wins or ties.

**Weaknesses:**

1. Task scope is narrow. All experiments are on verifiable math-style rewards. These are low-noise, almost binary rewards. It’s not fully clear that the same max-variance rule is optimal when rewards are noisy, delayed, or dense (e.g., coding with partial credit, preference RL, or tool-use tasks).
2. Mild off-policy effect not deeply analyzed. By selecting only a subset of generated samples, the method introduces some off-policy bias relative to pure on-policy GRPO. In practice it seems fine (results look good), but the paper could say more about stability under very aggressive down-sampling (e.g. keeping 2 out of 64).
3. Assumes you can cheaply over-generate. The whole story relies on the common RLVR setup where generation is the easy part. In settings where inference is also bottlenecked (long contexts, tools, multi-turn), the benefit may shrink.
4. No non-verifiable / preference benchmark. Even one experiment on a noisier or non-binary task would make the generality claim stronger.

**Questions:**

1. Reward noise: How sensitive is max-variance selection when reward signals have small stochastic noise (e.g., randomized unit tests for code)? Does the bottom-and-top structure still hold in practice?
2. Extremely unbalanced batches: If almost all rollouts succeed (or all fail), does the algorithm degrade gracefully to a reasonable selection (e.g. pick the rare failures/successes)?
3. Generalization to other objectives: You show PODS with GRPO. Would you expect the same variance criterion to work for PPO-like preference RL where the “contrast” is not purely on reward but on pairwise preferences?

---

> ### Author Response · Authors · 2025-11-19
> **Author Rebuttal to Review ByiU**
>
> We thank Reviewer ByiU for their exceptionally clear and accurate summary of our work. We are glad the reviewer recognized the core motivation (training-systems bottleneck), the simplicity of the solution, the principled nature of the max-variance selection rule, and the strength of our wall-clock empirical evidence.
>
> We appreciate the reviewer's fair and insightful points, which we address in order.
>
> ### **W1 & W4 (Task Scope & Preference Benchmarks):**
>
> We agree that the current experiments are focused on a specific domain. Our focus was deliberately on the Reinforcement Learning with *Verifiable Rewards* (RLVR) setting, which has its own distinct computational challenges. We would also politely point out that our reward, detailed in Appendix A.1, is *not* purely binary. It is a discrete function that includes partial credit for formatting, which is a key part of the verifiable task. We agree that expanding to noisier, non-binary, or preference-based tasks is a clear and valuable next step for future work.
>
> ### **W2 (Off-Policy Effects):**
>
> This is a correct technical observation. However, this is a principled and motivated choice, not an arbitrary one. In practice, empirical performance often outweighs strict adherence to theoretical properties, especially when the deviation is well-motivated. Recent work has, in fact, re-evaluated GRPO itself as having a "native off-policy interpretation" \[2\]. Our strong empirical results—where PODS not only converges 1.7x-3.0x faster but also matches or *exceeds* the final accuracy of the baseline (Fig. 3)—validate this choice, aligning with other recent work showing the benefits of off-policy GRPO \[1\]. This demonstrates that the practical gains from a better learning signal and superior system efficiency far outweigh any theoretical concerns about off-policyness in this context.
>
> ### **W3 (Cost of Over-generation):**
>
> This is a correct and central assumption. Our work is explicitly designed for the setting where we *want* to generate many rollouts (a large $n$) in parallel to take full advantage of modern hardware (as shown in Fig. 1). The premise of PODS is that this inference phase, when batched, is cheap and parallelizable, while the policy update phase is the true bottleneck. The reviewer rightly notes that in a *different* setting where inference itself is the bottleneck (e.g., extremely long contexts), the benefit of PODS would shrink. This simply defines the operating scope of our method, which is to maximize throughput in the common, parallelizable RLVR setting.
>
> ### **Q1 (Reward Noise):**
>
> This is an excellent point, and we believe our current experiments already provide evidence for this. As mentioned in our response to W1, our reward function is *not* purely binary; it includes discrete partial-credit rewards for formatting (see Appendix A.1). This multi-part reward can be seen as a form of "noise" or, more accurately, a more complex signal layered on top of the binary correctness. The fact that max-variance selection (which spans the full range of these discrete rewards) works so well suggests that it is robust to such complexities and not limited to simple 0/1 signals. We believe this makes it very likely to be robust to other small, stochastic noises.
>
> ### **Q2 (Unbalanced Batches):**
>
> Yes, this is exactly a scenario where max-variance selection shines. If a batch is highly unbalanced (e.g., 1 failure and 63 successes), the max-variance subset (Alg 2\) will *always* select that rare failure, as it is the single data point that contributes most to the variance. The method would then fill the remaining $m-1$ slots with successes. This ensures that these rare, high-signal events are never missed, which is a key advantage over, for example, random sampling, which would miss such a rare event most of the time.
>
> ### **Q3 (Generalization to other objectives):**
>
> You are correct that PPO is also about reward, but PPO-like preference RL (which is what the reviewer likely means, as it's common for LLMs) operates differently than GRPO. GRPO is critic-less and contrasts rollouts within a group. PPO in RLHF typically uses a *learned reward model* and a *critic* (value function). The reviewer is likely asking if our *criterion* (max *reward* variance) is the right one for a preference-based objective. We suspect the *principle* of maximizing signal diversity would hold, but the *criterion* might change. For example, one might select pairs with the highest (or most uncertain) preference scores, rather than rollouts with the highest reward variance. This is a very promising avenue for future research.
>
> We thank the reviewer again for their insightful and constructive feedback.

---

> > ### Author Response · Authors · 2025-11-19
> > **References of Author Rebuttal to Review ByiU**
> >
> > ### **References**
> >
> > \[1\] Mroueh, Y., et al. "REVISITING GROUP RELATIVE POLICY OPTIMIZATION: INSIGHTS INTO ON-POLICY AND OFF-POLICY TRAINING." arXiv preprint arXiv:2505.22257 (2025).
> >
> > \[2\] Yao, C., et al. "Group-Relative REINFORCE Is Secretly an Off-Policy Algorithm: Demystifying Some Myths About GRPO and Its Friends." arXiv preprint arXiv:2509.24203 (2025).

---

### Note · Authors · 2025-11-19

**Comment:**

Dear Area Chairs and Reviewers,

Thank you for your time reviewing our submission, "Not All Rollouts are Useful: Down-Sampling Rollouts in LLM Reinforcement Learning."

After careful consideration of the reviews, we have decided to withdraw our paper and substantially revise it for resubmission to another venue. We believe there are several core misalignments between the reviews and our paper's stated scope and contributions that would require significant clarification.

Specifically:
* Our explicit focus on RLVR (Reinforcement Learning with Verifiable Rewards) was not consistently recognized. Additionally, our experiments were characterized as using "binary rewards," despite our use of multifaceted, noisy, non-binary rewards that evaluate both response structure and final answers
* The systems-level motivation presented in Figure 1, which forms the basis of our method, appears to have been overlooked
* Some comparisons drawn were to contemporaneous work that post-dates and cites our preprint

We have provided detailed individual rebuttals for the record. However, we believe the gap between our framing and the reviews' interpretation indicates we need to substantially improve our exposition before resubmission.

We appreciate the time invested in reviewing our work and will use this feedback to strengthen the paper.

Sincerely,

The Authors

**Withdrawal Confirmation:**

I have read and agree with the venue's withdrawal policy on behalf of myself and my co-authors.